# First observations of sea ice flexural-gravity waves with ground-based radar interferometry in Utqiaġvik, Alaska

Dyre Oliver Dammann[1,2], Mark A. Johnson[3], Andrew R. Mahoney[1], Emily R. Fedders[1]

[1]Geophysical Institute, University of Alaska Fairbanks, Fairbanks, Alaska, Fairbanks, AK 99775, USA
[2]Norwegian Geotechnical Institute, 0806 Oslo, Norway
[3]College of Fisheries and Ocean Sciences, University of Alaska Fairbanks, Fairbanks, AK 99775, USA

*Correspondence to*: Dyre Oliver Dammann (dyre.dammann@ngi.no)

**Abstract.** We investigate the application of ground-based radar interferometry for measuring flexural-gravity waves in sea ice. We deployed a Gamma Portable Radar Interferometer (GPRI) on top of a grounded iceberg surrounded by landfast sea ice
near Utqiaġvik, Alaska. The GPRI collected 238 acquisitions in stare-mode during a period of moderate lateral ice motion during 23-24 Apr 2021. Individual 30-second interferograms exhibit ~20-50 s periodic motion indicative of propagating infragravity waves with ~1 mm amplitudes. Results include examples of onshore wave propagation at the speed predicted by the water depth and a possible edge wave along an ice discontinuity. Findings are supported through comparison with on-ice Ice Wave Rider (IWR) accelerometers and modeled wave propagation. These results suggest that the GPRI can be a valuable
tool to track wave propagation through sea ice and possibly detect changes in such properties across variable ice conditions.

## 1 Introduction

Ocean waves play an important role impacting the formation, dynamics, and break-up of sea ice as established by numerous studies (Squire et al., 1995; Squire, 2007). Waves in sea ice have gained increasing attention in recent years due to rapid loss of sea ice in the Arctic (Yadav et al., 2020) leading to enhanced fetch. This is expected to increase ocean wave activity and the
generation of swells which can penetrate far into the ice pack as flexural-gravity waves (Kohout et al., 2015). The propagation of infragravity waves through sea ice is complex as it depends upon resonant frequencies and can lead to leaky waves and edge waves along boundaries (Kovalev et al., 2020; Kovalev and Squire, 2020). Propagation of waves can in turn induce fracture, and break up ice floes into smaller pieces further accelerating sea ice decline (Thomson and Rogers, 2014).

The recognized significance of waves in ice and their dispersion and attenuation led to several advances in *in situ* and remote
sensing methods as well as multiple scientific experiments conducted from drifting sea ice (Squire, 2018). Early assessments of wave propagation in sea ice were carried out using wire strain gauges (Squire, 1978) and used to detect "ice coupled" flexural-gravity waves in landfast sea ice (Crocker and Wadhams, 1988). Tilt meters were later utilized with easier deployment

and maintenance (Czipott and Podney, 1989) partially through self-leveling mechanisms (Doble et al., 2006). Several other techniques have also proven valuable for wave detection in sea ice such as buoys, upward looking sonar (Thomson et al., 2019), and ship-based stereo imagery (Smith and Thomson, 2020).

Accelerometers are commonly utilized to measure waves in sea ice (Kohout et al., 2015; Sutherland and Rabault, 2016) and have significantly improved over the years (Doble et al., 2006) partly due to open source components (Rabault et al., 2020). In this work, we utilize a system named Ice Wave Rider (IWR) which is based on the VN100 inertial measurement unit (IMU) manufactured by Vectornav Co. This system measures 3d acceleration at 10 Hz with a three-axis accelerometer and a three-axis gyroscope. The components are enclosed in a Pelican Storm Case and can be strapped down to the ice for 60-day deployments with Iridium telemetry of data (Johnson et al., 2020).

Remote sensing approaches have also been used to evaluate waves including optical and radar altimetry (Collard et al., 2022) as well as lidar altimetry to evaluate waves in the marginal ice zone (Horvat et al., 2020). Synthetic aperture radar (SAR) has been used to estimate wave orbital velocity and wave height from backscatter distortion (Ardhuin et al., 2015; Ardhuin et al., 2017) and map wave fields through interferometry (Mahoney et al., 2016). These satellite-based approaches are valuable for evaluating waves in sea ice over large spatial scales, but are limited in temporal sampling. A higher sampling can be obtained with airborne systems (Sutherland et al., 2018), but logistics and cost can limit sampling to hours. For longer term observations of sea ice motion and deformation, a ground-based system can be a more practicable solution.

In a recent study, Dammann et al. (2021a) used a Gamma Portable Radar Interferometer (GPRI) stationed on floating sea ice to observe microscale horizontal strain. This demonstrated the ability of the GPRI to quantify and separate transient processes from a large-scale strain field and dynamically discriminate between regions of different properties. Additional work has been done to observe landfast sea ice from shore using a GPRI to discriminate stabilized zones and monitor ice movement in response to wind and current conditions (Dammann et al., in review). A key motivation for such work has been to investigate the potential for the GPRI system for seasonal monitoring of landfast ice and evolving stability due to changing ice and environmental conditions. This could help determine the application of the GPRI to detect conditions or dynamics as precursors to ice failure and breakout events such as horizontal strain and tidal displacement (Dammann et al., in review). However, an open question has been whether the GPRI could characterize waves in sea ice which together with long-term strain monitoring could help characterize ice conditions and impacts of waves on ice stability.

With only a limited GPRI dataset obtained from a grounded iceberg near Utqiaġvik, Alaska in April 2021, we demonstrate in this paper the application for monitoring mm-scale waves in landfast sea ice. First, we model the expected results from idealized harmonic waves and compare with GPRI observations. Second, we compare observations with ice displacement data derived from three IWRs deployed on the ice. Finally, we discuss wave properties, accuracy, and limitations due to secondary vertical and horizontal motion.

## 2 Data and methods

### 2.1 Ground-based radar interferometry of sea ice

In this work, we utilize the Gamma Portable Radar Interferometer (GPRI). This coherent radar system is capable of detecting mm-scale displacements in sea ice through interferometry i.e., the assessment of the phase change, $\Delta\Phi$, in the radar backscatter over time (Dammann et al., 2021a). $\Delta\Phi$ is proportional to the component of surface displacement aligned with the radar's line-of-sight (LOS), allowing relative horizontal displacement, $d_{hr}$, or relative vertical displacement, $d_{vr}$, to be calculated from the observed $\Delta\Phi$ as follows:

$$d_{hr} = \frac{\Delta\Phi \, \lambda_s}{4\pi \sin\theta} \, , \; d_{vr} = \frac{-\Delta\Phi \, \lambda_s}{4\pi \cos\theta} \tag{1,2}$$

depending on the signal wavelength, $\lambda_s = 0.017 \, m$ and incidence angle, $\theta$. For a GPRI system elevated above the ice surface, both horizontal and vertical motion can result in a significant LOS displacement (i.e., change in slant range) in the near field, within a few hundred meters of the GPRI. In this range, geometric constraints are required to resolve ambiguity between vertical and horizontal motions. Beyond such a distance, instrument sensitivity to vertical motion becomes negligible as incidence angle, $\theta$, approaches 90° and all phase change can be interpreted as horizontal.

The accuracy of the phase-derived ice motion depends on the interferometric coherence, a measure of how stable the reflected radar signal is over time, ranging between 0 (incoherent) and 1 (completely coherent). For points on the ice with high coherence, e.g., > 0.9, accuracy can reach sub-mm scale (Dammann et al., 2021a). However, accuracy can also be impacted by antenna movement (e.g., due to unstable GPRI footing) (Dammann et al., 2021a) or changing atmospheric conditions (Dammann et al., in review). The GPRI can be operated in either of two modes, scan mode, in which the antennas rotate while acquired data, or stare mode, where an image is acquired while looking in a fixed orientation. Here, we apply the stare mode, in which the GPRI collects continuous measurements in one direction at 100 Hz. The observations are interpreted as coming from a narrow (one-dimensional) strip, as the antenna generates a fixed fan beam spreading 0.4º in azimuth. Individual observation timespans were limited to 30 s. During these sub-minute windows, we expect atmospheric contributions to be negligible.

In stare-mode, interferograms are oriented in range-time space where each row represents the phase change since the start of the acquisitions. Each row thus represents cumulative phase change up until a particular time and each column represents a particular range point on the ice in the stare direction from the GPRI. We process all the interferograms by first averaging temporally to effectively reduce speckle and decrease the temporal sampling to 20 Hz. Then, we interpret the progressive $\Delta\Phi$ over 30 s as vertical displacement according to Equation 1, and subtract the mean displacement to easier identify the wave motion. We then subset the 30 s displacement timeseries based on low variability (RMSE < 0.3-0.5 mm compared to a 1 s running mean). The reduced sensitivity to vertical motion with range in combination with small ~ 1 mm observed waves we found it optimal to limit observations within 200 m of the GPRI and to areas with high coherence (>0.999) to ensure low noise in the observations.

## 2.2 Observations at Utqiaġvik

We carried out a series of 30 s long observations on the landfast sea ice near Utqiaġvik during April-May 2021. The landfast ice consisted of first year sea ice and incorporated a large iceberg grounded at 10 m depth ~2 km offshore. We stationed a GPRI on top of the iceberg ~6 m above sea level (Figure 1a). The radar alternated between staring in a direction across and along a ~200 m wide refrozen lead (cyan lines in Figure 1b) with a two minute lag repeated every ten minutes totaling 238 acquisitions. Within 200 m from the GPRI, we expect the vertical sensitivity to be sufficient to pick up vertical motion. Clear wave signals were only identified with the GPRI facing across the lead (solid cyan line) possibly due to the smooth, uniform ice conditions. We also deployed three Ice Wave Riders (IWRs) on the ice in the vicinity of the GPRI (green triangles in Figure 1a). These deployments enable a comparison between IWR- and GPRI-derived ice surface motion during only two across-lead acquisitions in which data overlapped (Table 1).

We evaluate ice thickness using measurements obtained between 16 Apr and 5 May 2021 as part of an annual community ice trail mapping project in Utqiaġvik. Ice thickness is derived from electromagnetic conductivity measurements obtained with snow machine pulling a GPS-equipped Geonics EM31-Mk2 ground conductivity meter (Druckenmiller et al., 2013). Measurements were collected every second at speeds < 5 m s$^{-1}$ ensuring ice thickness measurements no more than 5 m apart. Ice thickness ranged from 0.6 m to several meter thick with areas of smooth ice being ~1 m thick.

Table 1: Location and time of observations.

| Sensor | Start time (UTC) | End time (UTC) | Distance to IWR 33 (km) | Distance to GPRI (km) | Location (N,W) |
|---|---|---|---|---|---|
| GPRI | 00:56 Apr 23 | 00:44 Apr 24 | 1.4 | - | 71.361467, 156.630078 |
| IWR#33 | 00:19 Apr 24 | 01:05 May 26 | - | 1.4 | 71.374531, 156.631160 |
| IWR#34 | 00:30 Apr 24 | 17:20 May 22 | 0.6 | 1.2 | 71.371703, 156.616030 |
| IWR#35 | 23:46 Apr 23 | 23:14 Jun 2 | 2.4 | 1.0 | 71.353681, 156.618920 |

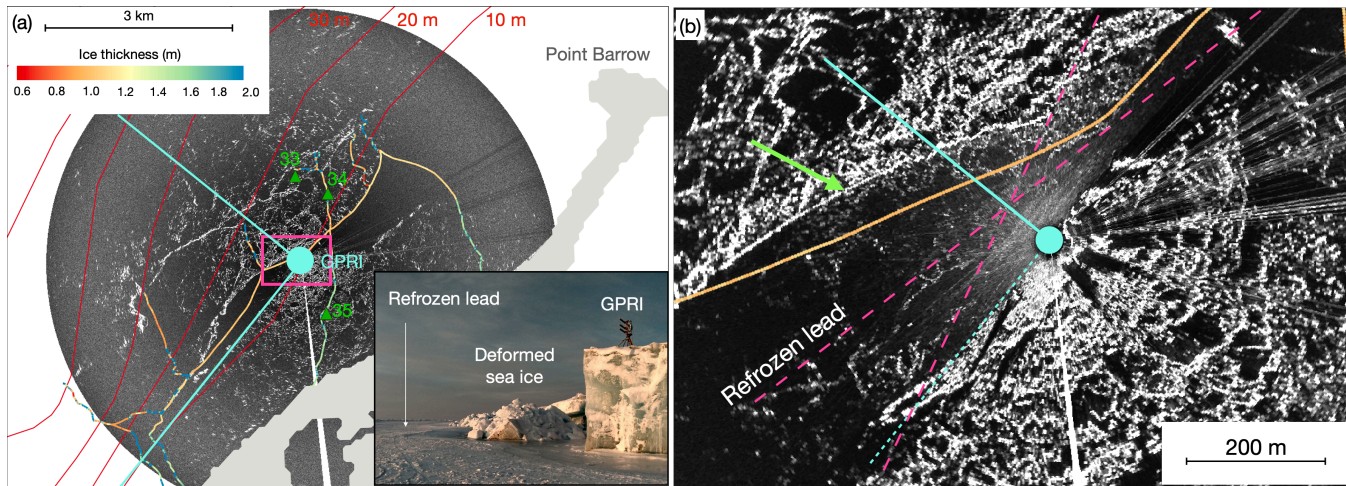

Figure 1: (a) Location of the GPRI superimposed on scan-mode backscatter imagery. The GPRI stared in the directions of the cyan lines. Bathymetry contours are drawn in red. Location of IWRs is identified with green triangles. Multi-colored line shows ice thickness as indicated by the color bar. Land is masked out in light gray. The pink rectangle indicates the location of the zoomed-in panel in (b). (b) Zoomed in area of the refrozen lead adjacent to the GPRI. The pink dashed lines indicate possible directions of wave propagation commented on in the text. The green arrow in (b) indicates deformed offshore edge of the refrozen lead.

## 2.3 Modeling wave dispersion and expected impact on stare-mode interferometry

We model vertical ice displacement, $d_v$, in response to harmonic waves to investigate how the GPRI can be used to detect and characterize waves in sea ice:

$$d_v(x, t) = A\sin(kx - \omega t + \varphi_0) \tag{3}$$

Where A is the amplitude, x is the distance from the GPRI, k is the wavenumber ($k = 2\pi/\lambda_w$ where $\lambda_w$ is the wavelength of the propagating wave), $\omega$ is the angular frequency ($\omega = -2\pi/T$), where T is the wave period), t is the time, and $\varphi_0$ is the phase of the surface wave at t = 0. If the GPRI is placed on a fixed location e.g., the shore or firmly grounded ice, the vertical ice surface displacement relative to the GPRI, $d_{vr_{fix}}$, will be equal and opposite the actual surface motion, that is:

$$d_{vr_{fix}} = -d_v. \tag{4}$$

We can model interferograms by substituting $d_{vr_{fix}}$ for $d_{vr}$ in Equation 2. To best illustrate this, we model waves with a period half of the observation window (T = 15 s) in water depth, H = 10 m (based on bathymetric contours in Figure 1) as an example. According to the shallow water approximation, the wave speed can be approximated to $c \approx 9.9$ m s[-1] through the following expression based on gravity, g, and water depth, H:

$$c = \frac{\lambda_w}{T} \approx \sqrt{gH} \tag{5}$$

We obtained the same speed, c, using the full dispersion equation (Johnson et al., 2021) and ~1 m ice thickness based on the EM31 survey. Through c, we can also approximate $\lambda_w$ = 150 m for T = 15 s. The resulting $d_{vr_{fix}}$ and associated interferogram is displayed in Figure 2a and b respectively. $\Delta\Phi$ is most significant in near range due to the decreasing sensitivity of the GPRI

135   to vertical motion with increasing range. $\Delta\Phi$ exhibits a periodic, diagonal pattern where we can identify both the wave period and observed speed, $c_o$, based on the pattern spacing and angle respectively (Figure 2b).

The speed observed with the GPRI, $c_o$, will differ from the speed of the wave, c, and depends upon the angle, $\alpha$, between the GPRI line of sight (LOS) and the propagation direction of the wave i.e., if $\alpha$ is non-zero, $c_0$ will be greater than $c$ because LOS distance between crests will be greater than the wavelength, $\lambda_w$:

140   $$c_o = \frac{\lambda_w}{T\cos\alpha} = \frac{c}{\cos\alpha} \qquad (6)$$

Incoming shallow water waves typically propagate perpendicular to bathymetric contour lines as wave speed decreases with decreasing depth (Equation 5). The LOS direction of the GPRI intersected the local isobaths approximately 15˚ from normal (Figure 1a), hence we expect to observe ~10 m s$^{-1}$ for waves propagating from offshore (modeled in Figure 2b). Inhomogeneities in the ice cover such as changing ice thickness, fractures, and rough ice can result in altered directionality of

145   wave propagation. As an example, $\alpha$ = 75º is expected to result in $c_o$ = 38 m s$^{-1}$ and the observed interferogram in Figure 2d. If the GRI is placed on floating ice (subject to vertical and horizontal motion due to sensor uplift and tilt) the phase patterns will be more challenging to interpret. This is further discussed in the appendix.

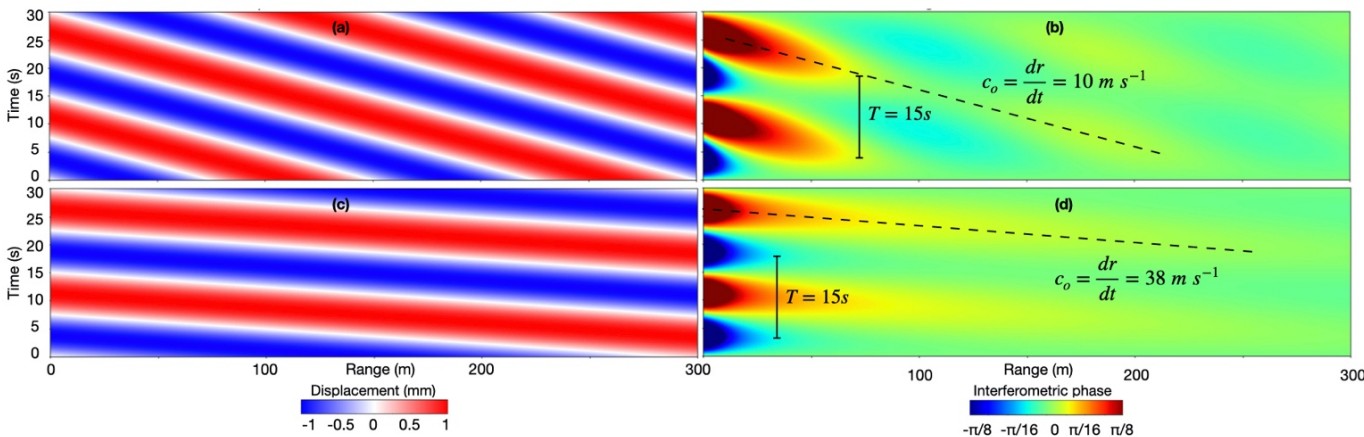

150   **Figure 2: (a) Relative vertical elevation (i.e., elevation difference between the GPRI and the ice surface) for simulated periodic oscillations in 10 m water depth (T = 15 s, A = 1 mm, $\lambda_w$ = 150 m) for LOS parallel to wave propagation and (b) associated synthetic interferogram. Panels (c) and (d) show results for the same simulated oscillations but with a LOS at 75˚ to the direction of propagation. The phase magnitudes of the patterns in (b) and (d) will differ based on the elevation of the GPRI system above the ice surface (here 6 m).**

155

## 2.4 Modeling superposition of infragravity waves and impact on stare-mode interferometry

During the season and in the region considered here around Utqiaġvik, Alaska, sea ice is widespread. From the dispersion relation the effects of sea ice on waves arise as products of the bending modulus and compressive stress terms times higher powers of the wavenumber (see Johnson et al. (2021) and references therein). These terms become small for the relatively long infragravity waves meaning ice has little effect. Sea swell on the other hand is significantly damped by the presence of sea ice (Bromirski et al., 2010). We have shown previously that infragravity waves can propagate over long distances across the ice-covered Arctic Ocean (Mahoney et al., 2016). As ocean swell is damped in an ice-covered sea, we assume that the wave forcing in this region is dominated by infragravity waves generated in the open ocean potentially hundreds of kilometers away (Bromirski et al., 2010; Bromirski et al., 2015). These waves are dispersive (Bromirski et al., 2010; Bromirski et al., 2015) so that the signals arriving in the vicinity of the GPRI are close to monochromatic during the short window of observations.

The modeled example (Equation 3 and Figure 2) demonstrates a single wave field. However, even with monochromatic infragravity waves, two wave fields can occur in sea ice when waves reflect off and propagates along inhomogeneities such as cracks and ridges (e.g., edge waves) (Marchenko and Semenov, 1994; Marchenko, 1999) and interact with the general wave field. Such reflected wave will have a similar period and speed, $c$, as the incoming wave field, but with different observed speed $c_o$. Hence, the resulting observed wave speed from individual observed waves $c_{o1}$ and $c_{o2}$ is $\bar{c_o} = (c_{o1} + c_{o2})/2$. In the case of a secondary wave field interfering, the superimposed waves can be expressed:

$$d_v(x, t) = 2A \sin\left(\bar{k}x - \bar{\omega}t\right) \cos\left(\frac{\Delta k}{2}x - \frac{\Delta \omega}{2}t\right) \tag{7}$$

Where $\bar{k}$ and $\bar{\omega}$ represents average values of the two waves. The phase speed of the combined wave field, represented by the first term in Equation (6b), is the average speed of the two waves. To illustrate this, we model a single infragravity wave field (Figure 3a) and superimpose an edge wave with $c_{o2} = 2c_{o1}$ (Figure 3b). We also model a standing wave as a result of a wave reflecting off a wall in the case the amplitude is conserved (Figure 3c). This is a potential scenario for waves interacting with the lead boundary/iceberg, but with uncertainties related to reflected amplitude and propagation angles.

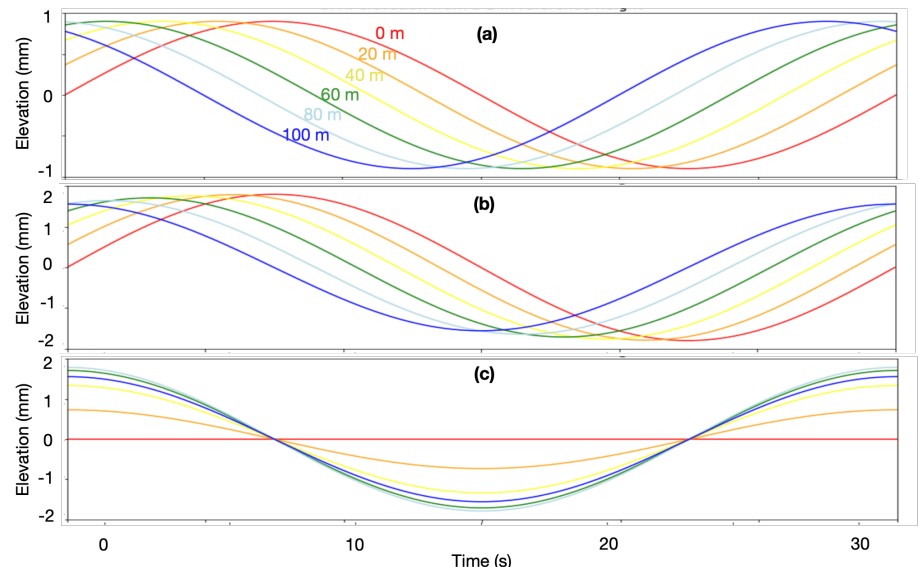

**Figure 3:** Modeled vertical displacement in the GPRI stare direction at different positions in range in the case of (a) a single wave field with $c_o$ =10 m s$^{-1}$, $\lambda$ = 300 m, A = 1 mm and T = 30 s, (b) the wave field in (a) superimposed on a wave field with the same A and T as in (a), but $c_o$ = 20 m s$^{-1}$ and $\lambda_o$ = 600 m, (c) a standing wave expected from an ideal reflection of the wave in (a). Colors represent different locations in range from 0 (red) to 100 m (blue).

### 2.5 Deriving wave properties from Ice Wave Rider data

We interpret the vertical acceleration from the IWRs in the form of the power spectral density. This is derived by partitioning the acceleration time-series into 15-minute segments with a 50% overlap and smoothing twice in frequency with a 1-2-1 weighting. We then display the amplitude-frequency distribution over time in the form of Welch periodograms. This enables the identification of ubiquitous bursts of activity typically less than 15 minutes (see example in Figure 4). We can estimate the predominate direction of wave propagation from the cross-correlation lag time, $l$, in between the acceleration signals measured at IWR#33 and IWR#34:

$$l = d \cos \psi / c \tag{8}$$

where $d = 630\ m$ is the distance in between the IWRs and $\psi$ is the propagation angle relative to the direct line in between the IWRs. We assume a similar wave speed as before of $c$ = 9.9 m s$^{-1}$ as the ice in between the sensors was mostly smooth and estimated to ~1 m thickness based on the nearby EM31 survey. A maximum correlation lag is thus expected to be $l = 64$ s ($\psi = 0$) and $l = 63$ s for waves propagating directly onshore ($\psi = 8°$). We derive the amplitude from the partitioned acceleration, $a_z$, as described by (Kohout et al., 2015) and further detailed in (Rabault et al., 2020):

$$A = a_z \omega^{-2} \tag{9}$$

We use a low frequency cutoff of T = 60 s, double the acquisition window of the GPRI. Lower frequency waves will thus be difficult to detect within the 30 s GPRI window and will start to resemble uniform sea level change.

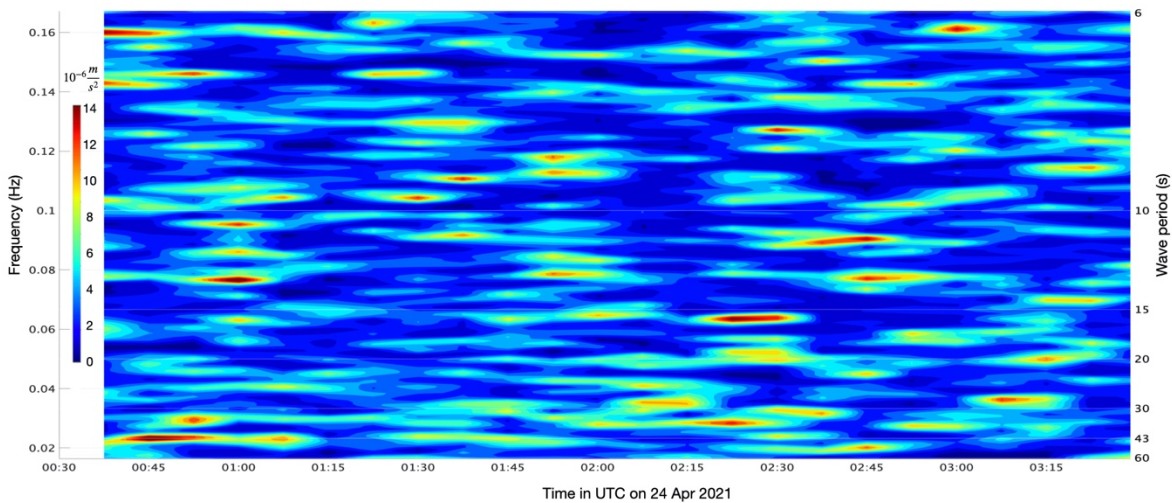

Figure 4: Example of vertical acceleration measured by IWR#33 during the first ~3 hours after deployment on 24 April 2021. The
acceleration is displayed as power spectral density in the frequency/time domain as a Welch periodogram. Numbers ranging from
6 to 60 to the right of the figure are period values in seconds corresponding with the frequency axis on the left-hand side

## 3 Results

### 3.1 Comparison of observations with a modeled wave

A good example of a surface wave-like signal detected by the GPRI is the interferogram acquired on 23 April at 21:56 UTC (referred to as E1) where a negative $\Delta\Phi$ signal (i.e., positive vertical displacement according to Equation 2) appears after the first ~15 seconds and propagates towards the GPRI over the last ~15 seconds (Figure 5a). To demonstrate the similarity with a wave signal, we model a wave according to Equation 3 from a fixed GPRI position. We model the wave with approximate wave properties based on the observations ($\lambda_w = 0.3\ km$, A = 0.9 mm, and T = 30 s). The similarity of interferometric phase patterns between the observations and model (Figure 5a and 5b respectively) confirm that the GPRI did not tilt significantly during acquisition and can therefore be treated as a fixed deployment. Although the single observed wave observed in Figure 5a matches close with the modeled wave field, it is worth noting that it lacks sign of the prior wave modeled as a second positive signal (red area in the bottom of Figure 5b). This suggests that although we observe an onshore wave, it may not be a part of a strict monochromatic wave field. Also, some vertical lines in Figure 5a differ significantly from surrounding lines and Figure 5b as they represent locations with low coherence.

Additionally, we identify individual range points on the ice with high coherence, which exhibit wavelike oscillations in displacement over time (Figure 6a). For each such point, we identify the time of maximum displacement and use this to track the progression in the wave crest and derive the speed, $c_o = 10$ m s$^{-1}$ with a standard error of 20 cm s$^{-1}$ (Figure 6b). The speed suggests that this wave likely propagated near directly onshore. The derived average amplitude for all high-coherence points

is $0.8 \pm 0.2\ mm$ (mean $\pm$ standard deviation), which appears representative for the amplitude beyond 80 m from the GPRI

225  (Figure 6c). However, as the wave approaches within 80 m of the GPRI, it increases slightly and then drops off closer than 35

m (dashed lines in Figure 6c).

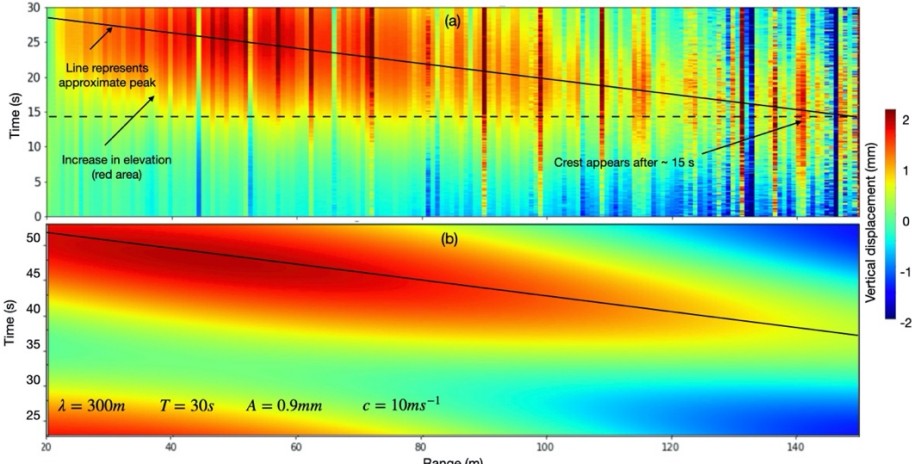

**Figure 5: (a) Phase-derived vertical displacement over 30 seconds at 21:56 UTC. The displacement is displayed in the range-time space along one fixed direction. Transient features are thus expected to have a diagonal nature, where the angle represents the**
230  **velocity of the feature. (b) Modeled displacement of a similar wave as in (a).**

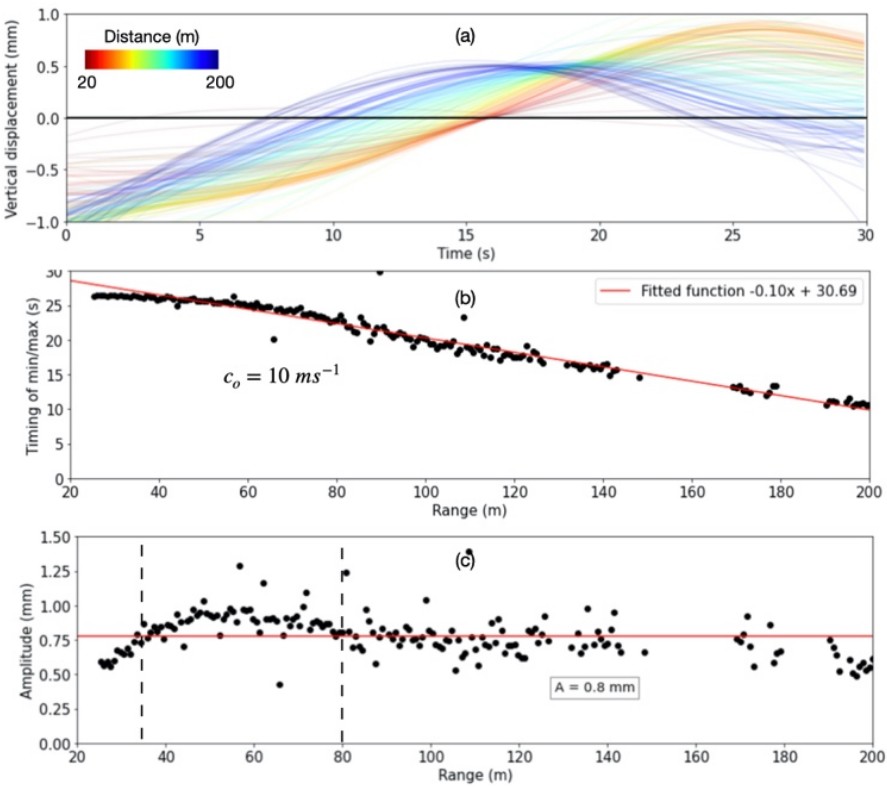

Figure 6: (a) Derived vertical displacement over 30 seconds on 23 April 2021 at 21:56 UTC (E1). Each line represents one location on the ice where warmer colors signify a closer distance to the radar. (b) Exact time of the wave crest at different distances from the radar. The linear fit indicates a wave velocity of 10.0 m s⁻¹. (c) Amplitude with range.

## 3.2 Observing superimposed wave fields validated with data from Ice Wave Riders

The three Ice Wave Riders (IWR) were continuously operating from approximately 00:30 UTC on April 24 (Table 1) overlapping two GPRI acquisitions in the offshore direction. The derived vertical displacement is thus suitable for validation of the GPRI data acquired at 00:32 and 00:44 UTC on 24 April. In this time window, we determine a cross-correlation lag time $l = 63.5\ s$ in between IWRs indicative of predominately onshore wave propagation (Equation 8). We further evaluate the displacement spectral characteristics in the Welch periodograms (Figure 7). Wave amplitude relates to frequency according to Equation 9 resulting in the lower frequencies in Figure 4 dominating the displacement (Figure 7). The displacement exhibits energetic signals between 30 and 60 s, well within the wave band for flexural waves, which persist for less than an hour. Both IWR#33 and IWR#34 suggest that several of the frequency peaks during and following the GPRI acquisitions (red line in Figure 7) are centered around 43 s (black lines in Figure 7). Although these peaks are small with an amplitude of ~1 cm, they are significantly above the derived noise floor as indicated in Figure 8. IWR#35 is situated behind grounded ice, in shallower water with thicker ice and does not exhibit the same signal as the other IWRs.

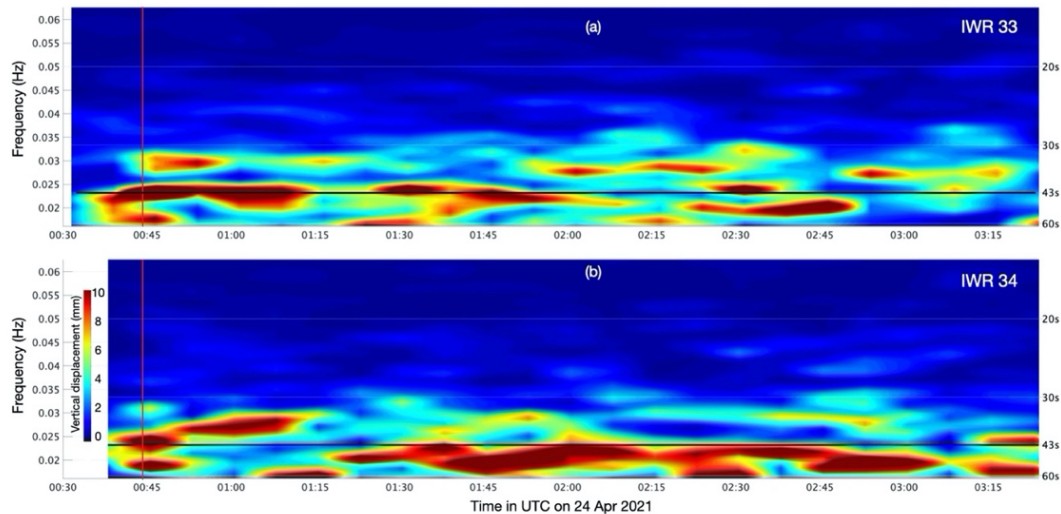

**Figure 7:** Spectra of the wave amplitude computed from the vertical acceleration from IWR#33 (Figure 4) and from IWR#34 presented as ~3-hour Welch plots from 24 April 2021. The displacement of IWR#33 is centered on a period of 43 s (black lines) at 00:44 UTC (time indicated with red vertical line).

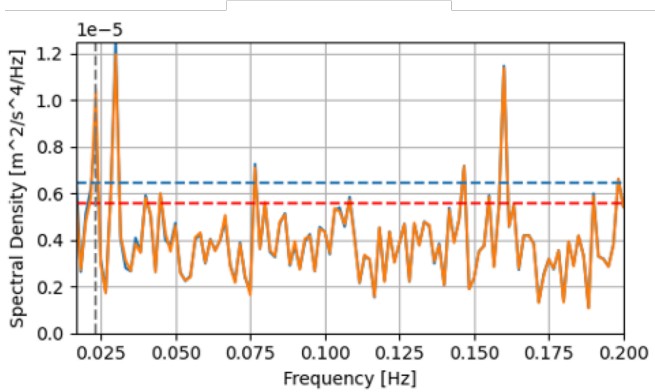

**Figure 8:** Solid lines indicate raw (blue) and ban-pass filtered (orange) spectral density during the first hour of the IWR record. Dashed red line indicate the noise floor of the IMU sensor (from VectorNav data sheet). Dashed blue line indicates noise floor derived from the IWR when resting on the floor. The 43 s wave signal at ~0.025 Hz exceeds both of these noise floors.

The last of the GPRI interferograms acquired at 00:44 UTC (referred to as E2) exhibits a clean wavelike motion (Figure 9a). We are able to track the timing of displacement maximum and minimums across most of the lead (Figure 9b) with a temporal lag of $21.4 \pm 0.5$ s. The period is exactly double at $T = 42.7 \pm 1.0$ s . This corresponds to the peak displacement frequency derived from IWR#33 (within measurement uncertainty) and suggests that the GPRI picks up the same wave field. The observed wave speed, derived from the slope of the maximum ice displacement (Figure 9b) is 27 m s$^{-1}$ (with a standard error of 25 cm s$^{-1}$) and larger than the speed of the incoming wave field at 10 m s$^{-1}$. We therefore assume that the observations

represent a reflected edge wave that can propagate along the lead superimposed on the incoming wave field identified with the IWRs. The actual speed of the two wave fields will be 10 m s$^{-1}$, but the observed speed of the edge wave $c_{o2}$ needs to be considered. The reflected wave will have conserved period, T, and the same for the superimposed waves as demonstrated in Figure 3b. As described by Equation 7, $\bar{c_o} = (c_{o1} + c_{o2})/2 = 27$ m s$^{-1}$ results in $c_{o2} = 44$ m s$^{-1}$ and suggests wave propagation at ~76° from the LOS. This is indicative of a edge waves traveling directly up the refrozen lead (pink dashed line in Figure 1b).

In addition to waves reflecting at ice boundaries and traversing along leads and ridges at an angle, we expect the incoming wave field to also be able to reflect directly off the iceberg with a conserved orientation. This is expected to lead to a standing wave as modeled in Figure 3c. A possible example of this was observed prior to the IWR data (Figure 10) where the minimum elevation of the trough occurs at ~15 s out to a node of 100 m and beyond at ~27.5 s, indicative of a standing wave with $T \approx$ 25 s. However, this signal bearing resemblance to a standing wave do not indicate more nodes (e.g., at ~300 m) and incorporates a linear trend that was removed for the analysis (dashed line in Figure 10), possibly due to a simultaneous increase in water level.

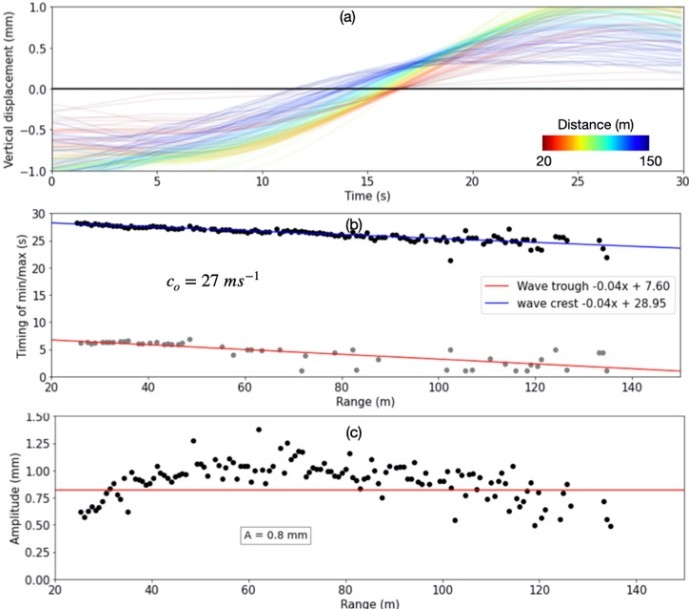

**Figure 9: (a) Derived vertical displacement over 30 seconds on 24 April 2021 at 00:44 UTC (E2). Each line represents one location on the ice where warmer colors signify a closer distance to the radar. (b) Exact time of the wave crest and troughs at different distances from the radar. The linear fit indicates a wave velocity of 27 m s$^{-1}$ and period of 43 s. (c) Amplitude with range.**

## 4 Discussion

### 4.1 Interpreting wave amplitude and speed

The wave amplitudes derived from GPRI stare-mode observations approximately 3 hours apart (E1 and E2) are similar to each other and peaks at ~60-80 m from the GPRI. Offshore from this distance, we attribute the increase in amplitude to shoaling as the water gets shallower. Closer than ~60 m from the GPRI, the wave amplitude drops by near 50% in both instances. A possible explanation for this is the presence of deformed, thicker ice near the GPRI (small picture in Figure 1a) and mechanical coupling between the floating sea ice and grounded iceberg.

The average observed speed in E1 matches well with the shallow water approximation indicative of an onshore wave. However, the speed appears to increase within ~70 m of the GPRI, apparent as a flattening of the curve representing the timing of the wave maximums (Figure 6b). One possible explanation is deflection as the wave approaches the grounded ice giving the appearance of higher speed due to a larger angle between propagation and LOS. Another possible explanation is that the ice in near-range is not in hydrostatic equilibrium, hence reaches its maximum value quicker than when the actual wave crest arrives. A third explanation is that the wave reflects off the grounded ice resulting in a maximum ice displacement which differs from the wave crest. The latter two explanations are based on the fact that the speed is strictly derived from the displacement maximum, which may not represent the wave crest and thus lead to inaccuracies.

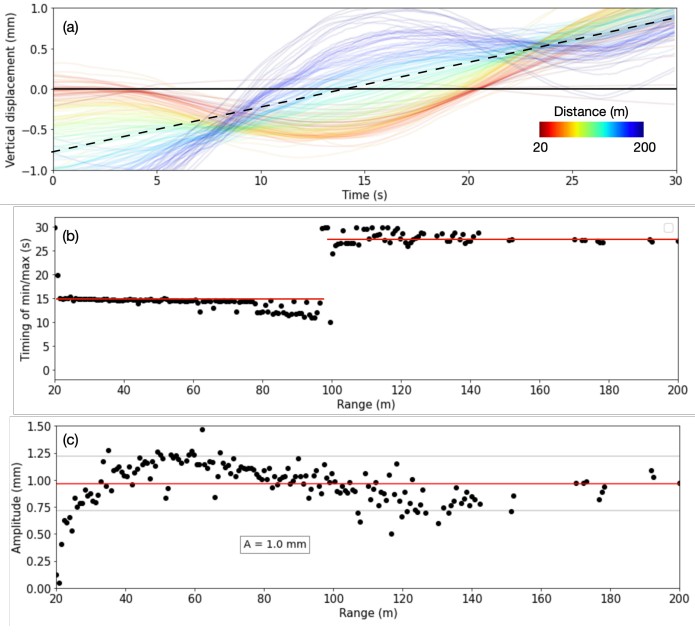

Figure 10: (a) Derived vertical displacement over 30 seconds on 23 April 2021 at 22:32 UTC (E2). Each line represents one location on the ice where warmer colors signify a closer distance to the radar. The dashed line indicates a linear trend removed for the analysis. (b) Exact time of the trough at different distances from the radar. The red lines indicate average values on each side of the node (c) Amplitude with range with red line indicating the mean.

## 4.2 Uncertainties related to propagation angle and amplitude

The average observed speed in E2 is 27 m s$^{-1}$, 17 m s$^{-1}$ higher than during E1. One possible explanation for this the presence of an edge wave traveling at an angle ~76˚ from LOS. Although we expect incoming waves to typically orient onshore, such waves can excite waves in ice discontinuities with a conserved period, which will propagate along such boundaries (Marchenko and Semenov, 1994; Marchenko, 1999; Evans and Porter, 2003). We speculate that E2 may incorporate one of these edge waves generated at the boundary between the refrozen lead and offshore ice (green arrow in Figure 1b). The wave then propagates along this boundary directly up the refrozen lead (pink dashed line in Figure 1b). A second explanation is the stark inhomogeneities in the ice, such as fractures, variable thickness, and rough ice leading to significant reorientation of the wave. InSAR-based snapshots of infragravity wave fields in sea ice indicate that waves fronts can be reorientated by tens of degrees by spatial variations in bathymetry and ice morphology (Mahoney et al., 2016).

The wave amplitude in E2 differs from what was observed with the IWRs by an order of magnitude. This is not necessarily surprising as the GPRI was not staring directly at any of the IWRs and satellite-based InSAR observations suggest significant variation in vertical motion (even featuring locations of zero motion) along a single wave front (Mahoney et al., 2016). Hence, in inhomogeneous ice, vertical displacement should be expected to significantly deviate from the average amplitude. Furthermore, if the E2 wave represents a generated edge wave, this may also have resulted in a diminished amplitude value in addition to attenuation and bathymetric influence on the amplitude. In essence, we do not observe nor expect the amplitude to be conserved in between the well-separated IWRs and the GPRI in the same way as the wave period. The derived amplitude is dependent upon the incidence angle, $\theta$, which is subject to uncertainties, predominately driven by inexact estimation of the antenna height atop of the iceberg. Uncertainties from sensor tilt due to iceberg motion are not considered as significant changes in $\theta$ could be identified in the antenna leveler and interferometric phase as illustrated in the Appendix.

## 4.3 Interpretation constraints due to multiple wave fields and horizontal motion

In addition to E1 and E2, many other examples exhibit wave-like motion in the GPRI data (see supplementary material), but can be challenging to interpret partly due to the presence wave fields with different sources and frequencies as well as horizontal motion. The IWR data suggest that essentially all frequencies in between 0.15-0.02 Hz can occur in the ice and multiple frequencies can be present at one time (Figure 4). The separation of multiple wave signals is challenging due to the short 30 s acquisition window resulting in the predominate capture of partial waves.

In addition to the interpretation challenges from multiple frequency signals, ice displacement in the horizontal plane appears to be the most common limitation for wave interpretation due to frequent horizontal ice motion. The GPRI is more sensitive to horizontal movement than to an equal displacement in the vertical. Hence, even modest lateral motion can complicate wave interpretation. This sensitivity may also enable the GPRI to potentially detect compressional or shear waves from ice-ice interaction propagating in the horizontal plane. However, such waves propagate at speeds on the order of kilometers per second (Rajan et al., 1993) and is expected to result in sharp peaks in displacement that may be difficult to detect. Even though horizontal movement often complicates interpretation, it can typically be identified in a phase signal. This is due to the low

vertical sensitivity with range that can lead to implausible values if interpreted as vertical (Dammann et al., 2021b) and the observed identical timing of displacement peaks with range.

## 5 Conclusions

This work leverages data from a 2021 coordinated GPRI/IWR campaign to demonstrate and validate the capture of flexural-gravity waves by the two sensors. The GPRI data was captured during modest wave activity <~ 1 mm and less than ~5% of acquisitions could be used to interpret wave properties due to the presence of what we interpret to be horizontal surface motion and interfering wave fields. We expect this percentage to be larger if data is acquired during larger-amplitude and more persistent wave activity. However, two particularly clear examples analyzed here demonstrate the ability to track waves with amplitudes of 1 mm and properties over a few hundred meters during the absence of secondary motion.

We also expect the collection of longer time series to aid interpretation in the future. In this work we were limited to 30 s of continuous radar acquisitions due to system constraints in stare-mode, specifically the data writing speed of the specific version of the GPRI hardware used here. This limitation does not extend to newer GPRI systems enabling continuous stare-mode acquisitions over hours to potentially days. This will open up possibilities for a more thorough evaluation of the wave signal. For instance, where we are here limited to look for single "clean" individual waves, we will be able to analyze combinations of multiple waves and frequencies for instance by applying Fourier analysis.

While the IWRs and similar on-ice installments enable the detection of waves and their properties, the GPRI provides the ability to spatially track individual waves. In addition to identifying wave period, speed, and amplitude, it can possibly help provide insight into how these properties change over a few hundred meters as a result of variations in the ice cover. Coordinated IWR/GPRI deployments can be particularly useful as it can resolve both regional variability in wave activity as well as the tracking of individual waves in select locations. Future deployments could attempt to place an IWR directly on the stare line of the GPRI to derive properties of the same wave with both sensors to investigate and resolve potential inconsistencies.

While the results discussed here show promise, the acquisition of longer time series and different types of waves is required to investigate potential applications. In this work, we analyzed infragravity waves, but the GPRI, with its high measurement frequency, can likely determine wave properties of ocean swells that can lead to ice fracture and destabilization. For ice subject to breakout, the GPRI is particularly valuable as it does not require deployment on the ice. Furthermore, the GPRI and the IWRs can provide near real-time data not available from other instruments (e.g., moorings) and can potentially aid in the development of an early warning system for ice breakout.

**Acknowledgements.** We thank Alexey Marchenko at the University Centre in Svalbard (UNIS) for valuable discussions on wave propagation in sea ice. We thank Matthew Druckenmiller at the National Snow and Ice Data Center (NSIDC) for providing ice thickness data. We are grateful to personnel at Ukpeaġvik Iñupiat Corporation (UIC) Science for logistical

support. We thank Alexey Marchenko again, Jim Thomson, and Marcel Kleinherenbrink for a thorough review which
significantly improved this paper.

**Funding.** This work supported by funding from the U.S. Office of Naval Research (award N000141912451) and the U.S. Army Corps of Engineers (award W913E521C0004)

**Author contribution.** D.O.D. conducted the interferometric processing and analysis and drafted the initial manuscript. M.A.J. contributed to all aspects of the analysis and manuscript. A.R.M and E.R.F. designed the field experiment, collected the data
and provided critical guidance on data interpretation. All co-authors provided valuable recommendations and corrections resulting in the final manuscript.

**Appendix**

Waves will impact $\Delta\Phi$ in different ways depending on whether the GPRI is stationed on moving ice or a fixed surface. In the case of a GPRI placed on floating ice that moves with the waves, the relative vertical displacement, $d_{vr_{float}}$, has an additional
component due to the vertical motion of the antenna:

$$d_{vr_{float}}(x, t) = -d_v + A\sin(-\omega t + \varphi_0) \tag{A1}$$

This second term is equivalent to $d_v$ evaluated at x=0 (red line in Figure A1a. As the waves propagate underneath the GPRI, the antenna will also be subjected to a variable tilt angle, $\varepsilon$, depending on the amplitude, A, and wavelength, $\lambda_w$:

$$\varepsilon = -\tan^{-1}\left(\frac{2\pi A}{\lambda_w}\right)\cos(-\omega t + \varphi_0) \tag{A2}$$

This will lead to periodic horizontal motion, $d_{hr_{float}}$ (blue line in Figure A1a), depending on the elevation of the GPRI antenna, $h$:

$$d_{hr_{float}} = -h\sin\varepsilon \tag{A3}$$

The resulting interferogram from both $d_{vr_{float}}$ and $d_{hr_{float}}$ (Figure A1b) has similarities to interferograms from a fixed system (Figure 2), but are more challenging to interpret (e.g., derive wave speed) due to the added complexity.

To highlight the contributions from vertical and horizontal motion to the phase for a floating GPRI, we isolate the relative vertical motion, $d_{vr_{float}}$, in Figure A2a. At distances equal to multiples of the wavelengths ($x = n\lambda_w$ ; $n \in \mathbb{N}$), the GPRI and the ice surface will move in phase leading to nodes where $d_{vr_{float}} = 0$ (Figure A2a). Halfway between these nodes ($x = (n + \frac{1}{2})\lambda_w$), the GPRI and ice surface will move out of phase leading to a maximum relative vertical displacement twice that of the wave amplitude. Similarly, we isolate the much smaller periodic horizontal motion, $d_{hr_{float}}$, component in (Figure
A2b). The resulting interferogram from $d_{vr_{float}}$ exhibits a predominate near-range phase response, while the interferogram

due to $d_{hr_{float}}$ exhibit a phase contribution nearly independent of range (Figure A2c and A2d respectively). The effect of Figure A2c and d results in the interferogram in Figure A1b.

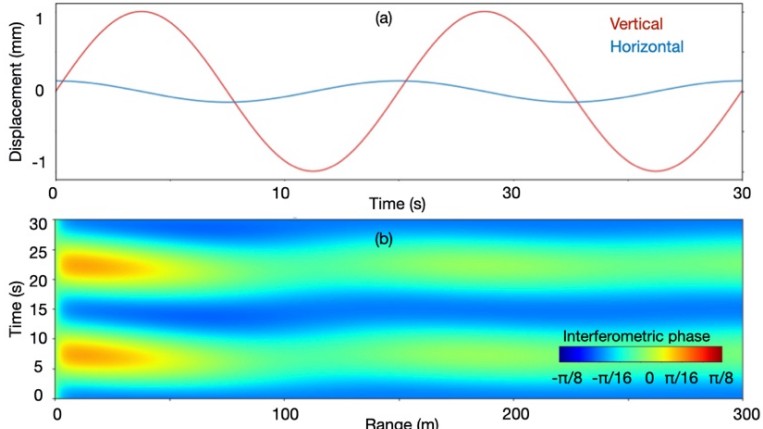

Figure A1: (a) Simulated vertical (red) and horizontal (blue) GPRI antenna motion as a result of waves (T= 15 s, A = 1 mm, $\lambda_w$ = 150 m, h = 2 m). (b) Simulated interferogram of a floating GPRI system based on a propagating wave including antenna motion in (a).

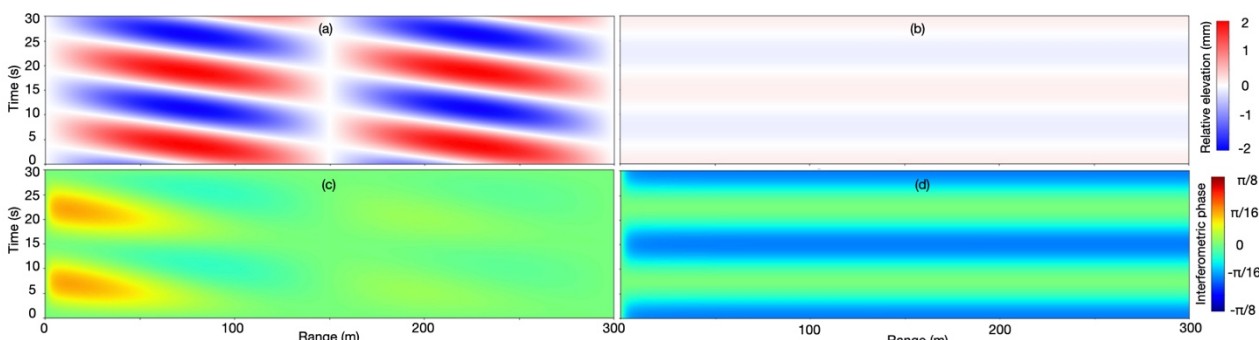

Figure A2: Relative change in vertical (a) and horizontal (b) distance between the GPRI antenna and the ice surface due to the wave in Figure A1. The resulting interferograms from the isolated motion in a and b (c and d respectively).

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
