# Peer review of "First observations of sea ice flexural-gravity waves with ground-based radar interferometry in Utqiagvik, Alaska"

_The Cryosphere, 2022_

## Referee Comment (RC1)

Review of the paper «Observing sea ice flexural-gravity waves with ground-based radar interferometry» by D.O. Dammann, M. A. Johnson, A.R. Mahoney, and E.R. Fedders

The paper is devoted to the description of the experience to observe waves propagating below floating ice with a Gamma Portable Radar Interferometer (GPRI). GPRI was used in the staring mode. According to the manual "very fast movements of millimetres to meters per second can be observed in this mode. The fast acquisition rate up to 4 kHz and pulse-to-pulse interferograms from a fixed antenna position (staring) can reveal rapid motion and vibrations." Therefore, the use of GPRI for the observations of flexural-gravity waves corresponds to the purpose of the unit. Sampling frequency of 100 Hz during 30 s allowed to reconstruct shape of ice surface deformed by waves along one of two staring directions. The water depth 10 m was measured in the region, and the ice thickness was investigated with EM-31. Wave periods measured with Ice Wave Riders were estimated above 30 s. Therefore, long wave approximation was used, and wave speed was estimated of about 10 m/s.

Sea ice in the region consisted of smooth ice of 1 m thickness in the frozen lead extended along the shore and rough ice around the lead with thickness changing from 0.6 m to several meters. The ice boundaries between the smooth lead ice and rough ice are curved. Length scale over which curvature changes are visible is 3 km. The GPRI was placed on the top of grounded iceberg 6 m above the water level. The iceberg was on the onshore boundary of the lead.

Two events (E1 and E2) of wave motion recorded with GPRI along offshore staring direction are discussed in the paper. In each event the wave speed along the specific direction was calculated by the comparing of wave phases in different points of the staring ray. The direction of wave propagation was calculated using assumption that the projection of measured speed on the direction of wave propagation equals 10 m/s. The GPRI data collected in E1 were interpreted as a record of wave form with amplitude about 0.9 mm propagating onshore with phase speed 10 m/s. Wave period and wavelength were estimated 30 s and 0.3 km. In E2 the GPRI data were compared with the accelerations recorded by two Ice Wave Riders (IWRs) deployed offshore the frozen lead. According to the GPRI and IWR data the period of observed wave motion was 43 s. The IWR data were interpreted as a record of wave form with amplitude about 10 mm propagating onshore with speed 10 m/s. The wavelength was 430 m. The GPRI data showed wave speed 27 m/s along the staring ray and amplitude of wave motion about 1 mm.

The processing of GPI data for the observations of harmonic wave is described in Sections 2.1 and 2.3. The influence of vertical motion of GPRI on the interpretation of harmonic wave records is discussed in the Appendix.

General comments to the paper are as follows.

The paper consists of theoretical part focused on the description of the methods of the data processing and experimental part describing organizing of the field works and obtained results. Performed measurements are new and interesting, but the theory doesn't explain observed features of wave motion. Discussion in the end of the paper is not specific enough. I recommend improving this gap.

Authors discuss that superposition of waves including waves reflected from the iceberg and waves reflected from the lead may influence the GPRI records. In this case wave motion in the region is not reduced to one harmonic wave. Therefore, methods described in Section 2.3 should be adjusted to the situation of wave superposition. Some simple demonstrations of possible effects would be useful to understand if wave superposition may explain the observations.

Incoming long waves may influence iceberg tilts and tilts of the GPRI. It leads to changes of the angle theta in equation (1.2). Analysis of this effect could be useful for the interpretation of the GPRI records.

Most strange result is that in E2 the GPRI records didn't show the wave propagating onshore which was registered with the IWRs. It is possible that incident onshore wave excites natural oscillations of ice in the lead propagating along the lead. But the incident wave should have an input in ice motions in the lead. Theoretical explanation of this result would benefit the paper.

Detailed comments.

Line 170. Reference on Table 2 is given in the text, but Table 2 is not presented in the text.

Line 170. $C_0=-10$ m/s. If it is speed, then it should be positive. The onshore direction is mentioned in the text.

IWR35 is listed in Table 1, but the information about records of IWR35 is not given in the text.

Line 195. $C_0=-27$. m/s ?. What means sign - ? The speed is positive/

The paper fits the journal profile and worth the publication after the revision. According to the comments above I recommend major revision.

---

## Author Comment (AC1)

Dear Dr. Marchenko

We greatly appreciate that you have taken the time to thoroughly read our manuscript and provide insightful and constructive suggestions. We have followed each of your recommendations which has helped improve and clarify the results and discussion around them.

Best regards,
Dyre Dammann

General comments to the paper are as follows.

The paper consists of theoretical part focused on the description of the methods of the data processing and experimental part describing organizing of the field works and obtained results. Performed measurements are new and interesting, but the theory doesn't explain observed features of wave motion. Discussion in the end of the paper is not specific enough. I recommend improving this gap.

We have now incorporated more discussion around superposition which now provides a more clear link between the modeled examples and previously incorporated results as well as newly included observations.

Authors discuss that superposition of waves including waves reflected from the iceberg and waves reflected from the lead may influence the GPRI records. In this case wave motion in the region is not reduced to one harmonic wave. Therefore, methods described in Section 2.3 should be adjusted to the situation of wave superposition. Some simple demonstrations of possible effects would be useful to understand if wave superposition may explain the observations.

This is a good suggestion and something that we now have included by incorporating superposition in the methods section in terms of an equation (Eq 7) and modeling examples (Figure 3), which do support the observations later on. We have provided a better explanation for the reflected edge wave as now discussed as a case of superposition supported by a simple model result. We have also incorporated a possible example of a standing wave to demonstrate another case of wave superposition (Figure 10) also modeled in Figure 3.

Incoming long waves may influence iceberg tilts and tilts of the GPRI. It leads to changes of the angle theta in equation (1.2). Analysis of this effect could be useful for the interpretation of the GPRI records.

We have now included a statement in Section 4.2 discussing uncertainties related to the angle theta. The largest uncertainties will be related to the antenna elevation and we think the simulations carried out in the Appendix are helpful in demonstrating potential effects in the case the iceberg were to move. We thus now write: "The derived amplitude is dependent upon the incidence angle, $\theta$, which is subject to uncertainties, predominately driven by inexact estimation of the antenna height atop of the iceberg. Uncertainties from sensor tilt due to iceberg motion are

not considered as significant changes in $\theta$ could be identified in the antenna leveler and interferometric phase as illustrated in the Appendix."

Most strange result is that in E2 the GPRI records didn't show the wave propagating onshore which was registered with the IWRs. It is possible that incident onshore wave excites natural oscillations of ice in the lead propagating along the lead. But the incident wave should have an input in ice motions in the lead. Theoretical explanation of this result would benefit the paper.

This is a great point. We naturally have to concider the incoming wave field as well. In response to your previous comment, we included a description of superposition and now aregue that E2 represents such a case of superposition of edge waves with the incoming wave field. Overall, this does not change the results much, but leads to a more reasonable explanation and also a predicted edge wave orientation directly parallel to the lead direction, which gives reason for strengthened confidence in this overall explanation.

Detailed comments.

Line 170. Reference on Table 2 is given in the text, but Table 2 is not presented in the text.

Good catch. The reference has been taken out

Line 170. $C\_0=-10$ m/s. If it is speed, then it should be positive. The onshore direction is mentioned in the text.

Changed.

IWR35 is listed in Table 1, but the information about records of IWR35 is not given in the text.

Now included: "IWR#35 is situated behind grounded ice, in shallower water with thicker ice and does not exhibit the same signal as the other IWRs."

Line 195. $C\_0=-27.$  m/s ?. What means sign - ? The speed is positive/

Good point. Sign has been removed

The paper fits the journal profile and worth the publication after the revision. According to the comments above I recommend major revision.

---

## Author Comment (AC2)

Dear Dr. Thomson,

Thank you so much for your time reviewing this paper and for your constructive comments and recommendations. We have made changes according to all your suggestions. Including measurements of expected noise in the IWR data was great hand has led to a stronger paper.

Best regards,
Dyre Dammann

This is a well-organized manuscript presenting exciting new observations of wave propagation in sea ice using Gamma Portable Radar Interferometer (GPRI). The results are mainly limited to two idealized examples from the larger GPRI dataset; these examples are both novel and convincing. Additional measurements from Ice Wave Recorders (IWRs) provide wave direction estimates that are essential to GPRI data interpretation, as well as ground-truth measurements for the waves inferred from the GPRI data.

The IWR measurements could use a slightly more careful treatment prior to final publication. Infragravity waves are notoriously difficult to measure with buoys (seafloor pressure gages are preferred). Certainly, the contrast between the broad-band signals of Fig 3 and the narrow-band (low frequency) signals of Figure 6 is striking... but what are the confidence levels for such small amplitude signals? I think it would be helpful to show an ensemble spectrum of the IWR observations from 24 Apr 2021 with confidence intervals based on the classic Chi^2 distribution for a spectrum of that many degrees of freedom. Something like Figure 7 from the recent Squire et al, Wave Motion, 2021. Does the infragravity peak at 43 sec stand-out above this confidence level? My guess is that it does, but it is not resolved particularly well.

Another way to improve the IWR measurements would be to present more details about the noise floor of the instrument. What does an ensemble spectrum from this instrument look like when place on land, away from industrial vibrations and other signals? Based on this empirical noise estimate, or on the sensor specifications, is it reasonable to expect to observe mm scale motions? These questions do not intend to undermine the results presented, but rather to provide better uncertainties estimates in comparing the GPRI and IWR results.

This is a great suggestion. We have now incorporated the noise floor both from the sensor manufacturer as well as estimated from when the IWRs are at rest. We have now included the following sentence: "Although these peaks are small with an amplitude of ~1 cm, they are significantly above the derived noise floor as indicated in Figure 7."

Some minor additional comments, by line:

line 21: This opening paragraph needs a sentence specific to infragravity waves in sea ice, which have many unique aspects (e.g., bound versus leaky modes, edge wave propagation) compared to sea and swell waves. Two recent papers by Kovalev (ECCS 2020 and CRST 2020) are worth citing.

Great suggestion. Included

line 28: a better reference for stereo measurements of waves in ice would be Smith and Thomson, Anal. of Glac. 2019.

Perfect, this is now changed

line 197: identical is a big word.   Suggest changing to "the same, within measurement uncertainty"

Agree. this has been changed

---

## Referee Report (RR1)

Review of "Observing sea ice flexural-gravity waves with ground-based radar interferometry" by Dammann et al.

General comments

The article describes the first application of GPRI measurements to study infragravity waves in sea-ice covered regions. The study is original and the suggested approach to detect infragravity waves is novel. The GPRI data shows potential to study infragravity waves in detail. The manuscript is clearly written and is well structured. I recommend it to be published in the Cryosphere after some modifications.

I have two concerns that should be addressed:
1) The authors should clarify why we need GPRI observations of infragravity waves. In the introduction only one line ("From a suitable … indefinitely.") is dedicated to this. Do we want to study ice-wave interactions, or spatiotemporally varying wave dynamics? In the conclusions it is suggested that sea-ice properties can be derived, but it is not clear if it is very useful on the spatial scale we are considering.
2) From the presented material in the paper alone in the paper I did not get fully convinced that we can derive infragravity wave properties from the GPRI. Of 238 records only two records showed a clear long-wave signal that matches the frequency of infragravity waves. I think figures of the other acquisitions should also be provided in supplementary material to give the reader a sense of their content. The first acquisition (E1) appears to have a nearly monochromatic signal of 30 s, but the data does not match 'part of the second crest' in figure 5. However, there is no IWR data available for this record. The IWR data for the stretch of three hours, however, shows that monochromatic waves hardly appear and typically there is some smearing or there are multipeak signals. So even though it appears we look at a monochromatic wave in E1, interference due to smeared spectrum might prevent the 'second crest' to show.  So, I wonder if the assumption of a monochromatic wave is valid.
For E2 there is IWR data available, but the phase velocity is different from expected. Several arguments are given for this discrepancy, but they appear to rely on rather strong assumptions (for example, reflected waves have amplitudes nearly equal to those of the inbound waves). Additionally, the amplitude differs an order of magnitude (in the IWR spectra of figure 7 a 10 mm vertical displacement is present, while the vertical displacement in figure 9 is only 1 mm).  The IWR33/34 data at the time of E2 also show two/three peaks in the spectrum. It would be nice to see a plot like 5a for record E2, and if possible, a model realization of 5b for E2 using the spectral information from the IWR33/34 as input (maybe using only three frequencies).
I needed the Mahoney et al. (2016) article to convince me that we see infragravity waves. I suggest to clearly state in the introduction that cm/mm-level infragravity waves have been observed in the Arctic near the considered region (Mahoney et al., 2016). State why this site is selected to study infragravity waves.  To convince the reader that the signals in the wave rider data are in fact infragravity waves, it should be supported with references to literature that show there are (regularly) ~0.02 Hz waves present in this area.

Technical corrections

Line 40: A reference to the review of Collard et al. (2022), "Wind-wave attenuation … " could be included.

Section 2.1: I think it is good to remind the reader that the GPRI is very directional. The azimuth footprint is several meters.

Line 78: The threshold for the coherence appears to be very strict. What is this threshold based upon?

Line 85: I assume that an acquisition is 30 seconds, like the evaluation length discussed in line 78.

Line 85: I wonder why the authors use the phrase 'every few minutes' and do not give a precise number. Is it operated manually?

Line 86: I guess this sentence refers to one of the cyan lines in the figure 1. Maybe it is good to indicate this in the figure and refer to it. As it is, the sentence can be read as if waves are only visible if they travel in the stare direction.

Line 125: I would rephrase this sentence. It practically always differs, so remove 'may differ … the wave, c, and,

Line 127: 'between crest is greater than'

Line 128: 'If the propagation'

Line 135: unit missing for alpha.

Line 125-135: I have the feeling a lot of words (and some repetition) are used to describe the geometric transformation with cos(alpha). This can be shortened.

Section 2.3 and elsewhere: While swell system have typically a very narrow angular spreads, (bound) infragravity waves have much larger angular spreads (Reniers and Zijlema, 2021). I am not aware how (free) infragravity waves propagate and evolve under sea ice. The authors should argue why using a model with one or two monochromatic waves suffices.

Section 2.3: Why do the authors give an example of waves within the swell regime, while the topic of the article is infragravity waves?

Line 172: Remove 'can'

Line 184-190: I feel figure 5 needs a more detailed description. I see several vertical stripes in figure 5a, which are not explained in the text. Secondly, a clear crest (peak) is visible along

the line, but the data (figure 5a) doesn't show the emergence of a second crest, which is visible in the model (figure 5a).

Line 230: "This corresponds to"

Line 140/235: The ratio of amplitudes between the reflected waves and the incoming waves are not considered. Is it valid to assume that they are equal? If not, the reflected wave might have quite different properties than estimated.

Line 260: Is there any reason to suspect the ice is not in hydrostatic equilibrium?

Line 263: This sentence is not completely clear.

Line 293: Something wrong with the sentence. I would also rephrase it, because it is suggestive. A 30 second integration time is too short to do a careful spectral analysis. For wave observations in the ocean integration over 10 minutes to 30 minutes is often used.

Line 311: ~1 mm wave propagation -> waves with amplitudes of 1 mm

---

## Referee Report (RR2)

Review of "Observing sea ice flexural-gravity waves with ground-based radar interferometry" by Dammann et al.

**General comments**

I think the authors handled my comments appropriately. I am looking forward to a longer continuous time span of measurements with a GRPI, because it seems promising. I only have one last comment and a few technical corrections.

**Specific comments**

In line 420 it is stated that more analysis is required to investigate potential applications like determining wave properties that can lead to fracture and destabilization. In the first paragraph of the introduction some properties of IG waves in ice are mentioned and that waves can induce fracture and break-up. However, I see no explicit reference that IG waves, considered in this paper, can cause fracture and break-up.

**Technical corrections**

Line 39-42: Rephrase these two sentences to include satellite radar altimetry (it is described in the Collard et al. (2022) paper as well).

Line 56: "we here" → "we demonstrate in this paper" or something comparable.

Line 98: "The observations are interpreted as coming from a narrow (one-dimensional) strip, as … azimuth" or something similar.

Line 105: "convert to…" something wrong with the sentence.

Line 107 and line 108/109: Subsetting on coherence appears to happen twice. Remove from one sentence.

Line 198: Here and on some other locations "ice-covered" like in the previous sentence.

Line 211: "We also model"

Line 271: Start with "Although…"

Line 384: "likely due in part to", resphrase

Line 415: "resulting in different wave propagation" feels like something is missing, please elaborate

Line 415: "particularly powerful", maybe tone down the sentence a bit: particularly useful or suitable

Line 417: "Here, furture deployments…" This sentence looks vague, rephrase. What does "here" refer to?

Line 420:  "determine wave frequencies and amplitudes" → "determine wave properties"

Supplementary material: units on the axis are missing

---

## Author Response (AR2)

Dear Dr. Kleinherenbrink,

Thank you so much for your helpful and positive recommendations. We have followed all your recommendations, which has resulted in a much improved paper. Please see below for detailed response to your suggestions.

Best regards,
Dyre Dammann

Review of "Observing sea ice flexural-gravity waves with ground-based radar interferometry" by Dammann et al.

General comments
The article describes the first application of GPRI measurements to study infragravity waves in sea-ice covered regions. The study is original and the suggested approach to detect infragravity waves is novel. The GPRI data shows potential to study infragravity waves in detail. The manuscript is clearly written and is well structured. I recommend it to be published in the Cryosphere after some modifications.

I have two concerns that should be addressed:

1) The authors should clarify why we need GPRI observations of infragravity waves. In the introduction only one line ("From a suitable … indefinitely.") is dedicated to this. Do we want to study ice-wave interactions, or spatiotemporally varying wave dynamics? In the conclusions it is suggested that sea-ice properties can be derived, but it is not clear if it is very useful on the spatial scale we are considering.

A great point. We have now clarified why we think this is valuable as part of prior efforts to evaluate the GPRI for landfast ice monitoring:

In a recent study, Dammann et al. (2021a) used a Gamma Portable Radar Interferometer (GPRI) stationed on floating sea ice to observe microscale horizontal strain. This demonstrated the ability of the GPRI to quantify and separate transient processes from a large-scale strain field and dynamically discriminate between regions of different properties. Additional work has been done to observe landfast sea ice from shore using a GPRI to discriminate stabilized zones and monitor ice movement in response to wind and current conditions (Dammann et al., in review). A key motivation for such work has been to investigate the potential for the GPRI system for seasonal monitoring of landfast ice and evolving stability due to changing ice and environmental conditions. This could help determine the application of the GPRI to detect conditions or dynamics as precursors to ice failure and breakout events such as horizontal strain and tidal displacement (Dammann et al., in review). However, an open question has been whether the GPRI could characterize waves in sea ice which together with long-term strain monitoring could help characterize ice conditions and impacts of waves on ice stability.

2) From the presented material in the paper alone in the paper I did not get fully convinced that we can derive infragravity wave properties from the GPRI. Of 238 records only two records showed a clear long-wave signal that matches the frequency of infragravity waves. I think figures of the other acquisitions should also be provided in supplementary material to give the reader a sense of their content.

All plots have now been included in the supplementary material and a mentioning of this data included in Section 4.3: "In addition to E1 and E2, many other examples exhibit wave-like motion in the GPRI data (see supplementary material), but can be challenging to interpret, likely due in part to the presence wave fields with different sources and frequencies as well as horizontal motion."

The first acquisition (E1) appears to have a nearly monochromatic signal of 30 s, but the data does not match 'part of the second crest' in figure 5. However, there is no IWR data available for this record. The IWR data for the stretch of three hours, however, shows that monochromatic waves hardly appear and typically there is some smearing or there are multipeak signals. So even though it appears we look at a monochromatic wave in E1, interference due to smeared spectrum might prevent the 'second crest' to show. So, I wonder if the assumption of a monochromatic wave is valid.

We agree. The title of the section has been changed as it implied the observations of a monochromatic wave field. The section heading now reads:  Comparison of observations with a modeled wave. We have also elaborated in the section: "It is worth noting that although the single observed wave observed in Figure 5a matches close with the modeled wave field, it lacks sign of the prior wave modeled as a second red area in the bottom of Figure 5b. This suggests that although we observe an onshore wave, it may not be a part of a strict monochromatic wave field."

For E2 there is IWR data available, but the phase velocity is different from expected. Several arguments are given for this discrepancy, but they appear to rely on rather strong assumptions (for example, reflected waves have amplitudes nearly equal to those of the inbound waves). Additionally, the amplitude differs an order of magnitude (in the IWR spectra of figure 7 a 10 mm vertical displacement is present, while the vertical displacement in figure 9 is only 1 mm). The IWR33/34 data at the time of E2 also show two/three peaks in the spectrum. It would be nice to see a plot like 5a for record E2, and if possible, a model realization of 5b for E2 using the spectral information from the IWR33/34 as input (maybe using only three frequencies).

Figure 5 is mostly added to show an initial observed interferogram and the full wave signal. This example makes for an interesting comparison as the wave speed is only 10 m s-1 so that the speed can be easily visualized and tracked. Example E2 look more horizontal due to the much

higher speed and is more difficult to use to determine speed. Below is an example of a modeled wave with three frequencies 0.024, 0.017, and 0.03 Hz. A fair comparison is more challenging than in Figure 5 due to the unknown orientation and relative phase of the individual waves with each frequency.

[Figure]

I needed the Mahoney et al. (2016) article to convince me that we see infragravity waves. I suggest to clearly state in the introduction that cm/mm-level infragravity waves have been observed in the Arctic near the considered region (Mahoney et al., 2016). State why this site is selected to study infragravity waves. To convince the reader that the signals in the wave rider data are in fact infragravity waves, it should be supported with references to literature that show there are (regularly) ~0.02 Hz waves present in this area.

This has now been included in a new subsection 2.4

Technical corrections
Line 40: A reference to the review of Collard et al. (2022), "Wind-wave attenuation … " could be included.

Done

Section 2.1: I think it is good to remind the reader that the GPRI is very directional. The azimuth footprint is several meters.

This has been clarified: "This limits observations to a single line as the antenna generates a fixed fan beam spreading 0.4º in azimuth"

Line 78: The threshold for the coherence appears to be very strict. What is this threshold based upon?

This has now been clarified: "We then subset the 30 s displacement timeseries based on low variability (RMSE < 0.3-0.5 mm compared to a 1 s running mean) as well as coherence. The reduced sensitivity to vertical motion with range in combination with small ~ 1 mm observed

waves we found it optimal to limit observations to areas with high coherence (>0.999) to ensure low noise in the observations."

Line 85: I assume that an acquisition is 30 seconds, like the evaluation length discussed in line 78.

Absolutely. This has now been stated.

Line 85: I wonder why the authors use the phrase 'every few minutes' and do not give a precise number. Is it operated manually?

Agree, this was not very precise. Now improved clarity by stating: "The radar alternated between staring in a direction across and along a ~200 m wide refrozen lead (cyan lines in Figure 1b) with a two minute lag repeated every ten minutes"

Line 86: I guess this sentence refers to one of the cyan lines in the figure 1. Maybe it is good to indicate this in the figure and refer to it. As it is, the sentence can be read as if waves are only visible if they travel in the stare direction.

This has now been done by using only a solid color for the across-lead direction and clarifying this: "Clear wave signals were only identified with the GPRI facing across the lead (solid cyan line) possibly due to the smooth, uniform ice conditions."

Line 125: I would rephrase this sentence. It practically always differs, so remove 'may differ … the wave, c, and,

Done

Line 127: 'between crest is greater than'

Done

Line 128: 'If the propagation'

Agree, that is better. Done

Line 135: unit missing for alpha.

Added

Line 125-135: I have the feeling a lot of words (and some repetition) are used to describe the geometric transformation with cos(alpha). This can be shortened.

This has now been significantly shortened

Section 2.3 and elsewhere: While swell system have typically a very narrow angular spreads,

(bound) infragravity waves have much larger angular spreads (Reniers and Zijlema, 2021). I am not aware how (free) infragravity waves propagate and evolve under sea ice. The authors should argue why using a model with one or two monochromatic waves suffices.

We have now created an additional subsection, now Section 2.4 where we both justify the assumption of monochromatic nature of infragravity waves and also adjusted the model according to infragravity waves.

Section 2.3: Why do the authors give an example of waves within the swell regime, while the topic of the article is infragravity waves?

It was a bit easier to see, but we realize that may be confusing. This has now been changed in the plot.

Line 172: Remove 'can'

Done

Line 184-190: I feel figure 5 needs a more detailed description. I see several vertical stripes in figure 5a, which are not explained in the text. Secondly, a clear crest (peak) is visible along the line, but the data (figure 5a) doesn't show the emergence of a second crest, which is visible in the model (figure 5a).
This has now been clarified: "It is worth noting that although the single observed wave observed in Figure 5a matches close with the modeled wave field, it lacks sign of the prior wave modeled as a second red area in the bottom of Figure 5b. This suggests that although we observe an onshore wave, it may not be a part of a strict monochromatic wave field. Also, some vertical lines in Figure 5a differ significantly from surrounding lines and Figure 5b as they represent locations with low coherence."

Line 230: "This corresponds to"

Done

Line 140/235: The ratio of amplitudes between the reflected waves and the incoming waves are not considered. Is it valid to assume that they are equal? If not, the reflected wave might have quite different properties than estimated.

Wave reflection at an ideal wall conserves wave amplitude and we assume a reflected to incident amplitude ratio of 1 simply to illustrate our case. We have clarified this now by stating: "We are also modeling a standing wave as a result of a wave reflecting off a wall in the case the amplitude is conserved (Figure 3c). This is a potential scenario for waves interacting with the lead boundary/iceberg, but with uncertainties related to reflected amplitude and propagation angles."

Line 260: Is there any reason to suspect the ice is not in hydrostatic equilibrium?

Ice near and attached to grounded ridges can be held down for instance at high tide. This can sometimes be clearly observed when water comes up auger hole.

Line 263: This sentence is not completely clear.

This has now been clarified

Line 293: Something wrong with the sentence. I would also rephrase it, because it is suggestive. A 30 second integration time is too short to do a careful spectral analysis. For wave observations in the ocean integration over 10 minutes to 30 minutes is often used.

This refers to the IWR data, which considers a longer timespan.

Line 311: ~1 mm wave propagation -> waves with amplitudes of 1 mm

Changed

---

## Author Response (AR3)

Dear Dr. Kleinherenbrink,

Thank you so much for once again looking over our paper so thoroughly and providing great comments. We have followed your recommendations which has lead to a better manuscript.

Best regards,
Dyre

Review of "Observing sea ice flexural-gravity waves with ground-based radar interferometry" by Dammann et al.

General comments: I think the authors handled my comments appropriately. I am looking forward to a longer continuous time span of measurements with a GRPI, because it seems promising. I only have one last comment and a few technical corrections.

We agree, we are looking forward to aquiring more data as well

Specific comments In line 420 it is stated that more analysis is required to investigate potential applications like determining wave properties that can lead to fracture and destabilization. In the first paragraph of the introduction some properties of IG waves in ice are mentioned and that waves can induce fracture and break-up. However, I see no explicit reference that IG waves, considered in this paper, can cause fracture and break-up.

Although we analyze infragravity waves in this paper, we suggest that the GPRI-approach can likely determine wave properties of swells as well. We have now clarified this in the conclusions by stating: "While the results discussed here show promise, the acquisition of longer time series and different types of waves is required to investigate potential applications. In this work, we analyzed infragravity waves, but the GPRI, with its high measurement frequency, can likely determine wave frequencies and amplitudes of ocean swells that can lead to ice fracture and destabilization."

Technical corrections

Line 39-42: Rephrase these two sentences to include satellite radar altimetry (it is described in the Collard et al. (2022) paper as well).

Good point. Done

Line 56: "we here" à "we demonstrate in this paper" or something comparable.

Done

Line 98: "The observations are interpreted as coming from a narrow (one-dimensional) strip, as … azimuth" or something similar.

Agree, that is more correctly stated. Done

Line 105: "convert to…" something wrong with the sentence.

This has been changed to: ". Then, we interpret the progressive $\Delta\Phi$ over 30 s, convert to vertical displacement according to Equation 1…"

Line 107 and line 108/109: Subsetting on coherence appears to happen twice. Remove from one sentence.

Good catch. Removed one

Line 198: Here and on some other locations "ice-covered" like in the previous sentence.

Done

Line 211: "We also model"

Done

Line 271: Start with "Although…"

Done

Line 384: "likely due in part to", resphrase

Done

Line 415: "resulting in different wave propagation" feels like something is missing, please elaborate

Definitely. This has now been changed to: "In addition to identifying wave period, speed, and amplitude, it can possibly help provide insight into how these properties change over a few hundred meters as a result of variations in the ice cover."

Line 415: "particularly powerful", maybe tone down the sentence a bit: particularly useful or suitable

Done

 Line 417: "Here, furture deployments…" This sentence looks vague, rephrase. What does "here" refer to?

Removed "here" as unnecessary

Line 420: "determine wave frequencies and amplitudes" à "determine wave properties"

Done

Supplementary material: units on the axis are missing

Added